# The Role of Soluble Adenylyl Cyclase in the Regulation of Flagellar Motility in Ascidian Sperm

**DOI:** 10.3390/biom13111594

**Published:** 2023-10-30

**Authors:** Kogiku Shiba, Kazuo Inaba

**Affiliations:** Shimoda Marine Research Center, University of Tsukuba, Shimoda 415-0025, Japan; kinaba@shimoda.tsukuba.ac.jp

**Keywords:** calcium, sperm motility, cilia, protein kinase

## Abstract

Flagellar motility in sperm is activated and regulated by factors related to the eggs at fertilization. In the ascidian *Ciona intestinalis*, a sulfated steroid called the SAAF (sperm activating and attracting factor) induces both sperm motility activation and chemotaxis. Cyclic AMP (cAMP) is one of the most important intracellular factors in the sperm signaling pathway. Adenylyl cyclase (AC) is the key enzyme that synthesizes cAMP at the onset of the signaling pathway in all cellular functions. We previously reported that both transmembrane AC (tmAC) and soluble AC (sAC) play important roles in sperm motility in *Ciona*. The tmAC plays a major role in the SAAF-induced activation of sperm motility. On the other hand, sAC is involved in the regulation of flagellar beat frequency and the Ca^2+^-dependent chemotactic movement of sperm. In this study, we focused on the role of sAC in the regulation of flagellar motility in *Ciona* sperm chemotaxis. The immunochemical analysis revealed that several isoforms of sAC protein were expressed in *Ciona* sperm, as reported in mammals and sea urchins. We demonstrated that sAC inhibition caused strong and transient asymmetrization during the chemotactic turn, and then sperm failed to turn toward the SAAF. In addition, real-time Ca^2+^ imaging in sperm flagella revealed that sAC inhibition induced an excessive and prolonged Ca^2+^ influx to flagella. These results indicate that sAC plays a key role in sperm chemotaxis by regulating the clearance of [Ca^2+^]_i_ and by modulating Ca^2+^-dependent flagellar waveform conversion.

## 1. Introduction

Sperm flagellar motility is crucial for fertilization success among most sexual organisms [1]. Spermatozoa sense environmental cues then regulate motility to reliably reach the egg of the same species. Soluble adenylyl cyclase (sAC) is an evolutionally conserved enzyme, which synthesizes cyclic AMP (cAMP) and plays an important role for the regulation of sperm motility from invertebrates to mammals [2,3]. It has been reported that sAC is regulated by calcium [4] and bicarbonate [5], and not by G protein subunits, which are known regulators of transmembrane AC (tmAC) [6]. Sea urchin sAC is localized in both the head and flagella of sperm and participates in both the acrosome reaction and motility [7]. Mammalian sAC is involved in sperm motility activation and capacitation [8,9,10] and is required for male fertility. A sAC inhibitor is proposed as an effective on-demand contraceptive for men [11]. The expression of multiple sAC isoforms has been reported, and truncated sAC containing two cyclase catalytic domains is dominant in mature sperm in mammals, suggesting that it is responsible for the regulatory mechanism in sAC functions [12].

We previously reported that both sAC and tmAC are present and regulate sperm motility in the ascidian *Ciona intestinalis* [13]. The tmAC plays a major role in the initial activation process in sperm motility. On the other hand, sAC is involved in the regulation of the flagellar beat frequency and the Ca^2+^-dependent chemotactic movement of sperm. Motility activation and chemotaxis of *Ciona* sperm are induced by the same molecule, called the SAAF (sperm activating and attracting factor) [14]. In the sperm signaling pathway triggered by the SAAF, cAMP is involved in both the signal transduction through cyclic nucleotide-regulated channels (CNR) and the regulation of flagellar motors through protein phosphorylation of axonemal proteins in a cAMP-dependent manner [15,16]. Four types of CNR, including two CNG (cyclic nucleotide-gated) channels, Ci-tetra KCNG (tetrameric, cyclic nucleotide-gated, K+-selective) and Ci-HCN (hyperpolarization-activated and cyclic nucleotide-gated), are expressed in *Ciona* sperm and localized in sperm flagella [17,18]. The HCN channel is also suggested to regulate [Ca^2+^]_i_ concentration and sperm motility during chemotaxis [17]. cAMP-dependent phosphorylation of the subunits of dynein motor proteins, a 21 kDa light chain of outer arm dynein (LC2) and a 26 kDa axonemal protein have been reported to be present when *Ciona* sperm activates flagellar motility with the SAAF [15].

Here, we examined the role of sAC in the regulation of flagellar motility in *Ciona* sperm chemotaxis. The immunochemical analysis revealed that several isoforms of sACs were expressed and localized in *Ciona* sperm flagella. Furthermore, we analyzed sperm flagellar motility and Ca^2+^ dynamics during chemotaxis in the presence of a sAC inhibitor. Our results suggest that sAC plays a key role in sperm chemotaxis by regulating the clearance of [Ca^2+^]_i_ and by modulating Ca^2+^-dependent flagellar waveform conversion.

## 2. Materials and Methods

### 2.1. Materials

The ascidian *C. intestinalis* (type A; also called *C. robusta*) was collected from Onagawa Bay near the Onagawa Field Research Center, Tohoku University, or obtained from the National BioResource Project for *Ciona* (https://marinebio.nbrp.jp, accessed on 27 October 2023). Animals were kept in aquaria under constant light for accumulation of gametes without spontaneous spawning. Semen samples were collected by dissecting the sperm duct and kept on ice until use.

### 2.2. Chemical and Solutions

Artificial seawater (ASW) was composed of 462.01 mM NaCl, 9.39 mM KCl, 10.81 mM CaCl_2_, 48.27 mM MgCl_2_ and 10 mM Hepes-NaOH (pH 8.0). Ca^2+^-free seawater (CaFSW) was as ASW without CaCl_2_ but with 478.2 mM NaCl and 2.5 or 10 mM EGTA. Inhibitors (KH-7) from Chemical Diversity Research Institute (Khimki, Russia) were dissolved in dimethyl sulfoxide (DMSO) and added to an appropriate concentration in ASW. Other reagents were of analytical grades. SAAF was synthesized as described previously [19,20].

### 2.3. Preparation of Antibodies and Immunoblotting

Polyclonal antibodies were raised against two cites (sAC-C1 and sAC-PL, Figure 1A) of sAC in mice as previously performed [17]. The following PCR primers were used for amplification of the open reading frame for sAC-C1 [5′-GCGCGGATCCATCGGTGGAGCAAATAAGCC-3′ (sense) and 5′-GCGCGAATTCCTAACGATCACGAGGACTCA-3′ (antisense)], and sAC-PL [5′-GCGCGGATCCTTGGGTAAAGAACGAAGAGT-3′ (sense) and 5′-GCGCCTCGAGCTACAATCTCCCCGACTGA-3′ (antisense)]. The PCR products were subcloned into a pET28a vector and transfected into *Escherichia coli* BL21(DE3). Protein expression was induced using 0.5 mM IPTG (isopropyl β-D-thiogalactoside). The recombinant proteins were purified using Chelating Sepharose Fast Flow (GE Healthcare, Chicago, IL, USA) and injected into mice four times with 10-day intervals between each injection. Antiserum was collected from mice and kept in aliquots at −80 °C. To prepare Triton X-100 soluble sperm protein, semen were suspended in 20 volumes of CaFSW and centrifuged at 2000× *g* for 5 min at 4 °C. The resulting pellet was suspended in an extraction medium containing 1% Triton X-100, 0.2 mM phenylmethylsulfonyl fluoride (PMSF), 0.2 mM dithiothreithol (DTT) and 10 mM Tris-HCl (pH 8.0). The suspension was centrifuged at 17,900× *g* for 20 min at 4 °C. The supernatant was mixed in a ratio of 4:1 with the sample buffer: 62.4 mM Tris-HCl, 2% SDS, 4% glycerol, 0.004% bromophenol-blue, pH 6.8 and boiled at 37 °C for 20 min. Proteins were separated using SDS-PAGE and transferred to polyvinylidene difluoride membranes. Membranes were treated with 7.5% skim milk in PBST (PBS containing 0.05% Tween 20) to prevent non-specific protein binding. Blots were incubated with anti-Ci-sAC-C1 (1:1000) or anti-Ci-sAC-PL (1:1000) primary antibodies for 1 h at room temperature. After washing them with PBST three times, blots were incubated with HRP-conjugated secondary antibodies at 1:10,000 for 30 min at room temperature. After washing them with PBST three times, immunoreactive bands were detected using ECL-prime (GE Healthcare). Signals were detected using the LAS-4000 mini-imager (Fujifilm, Tokyo, Japan).

### 2.4. Immunofluorescence Microscopy

Immunofluorescence microscopy was performed as outlined in a previous study [17], with slight modifications. Semen were suspended in 50 volumes of CaFSW with 1 mM theophylline for motility activation. Sperm were attached on slides pre-coated with 1 mg/mL poly-l-lysine. After incubation for 5 min, the sperm were fixed with methanol at −30 °C for 10 min and rehydrated with excess volume of PBS. After washing with PBS, the slides were incubated with blocking buffer (10% goat serum in PBS) for 1 h in a moist chamber, followed by incubation with mouse primary antibody against Ci-sAC-C1, Ci-sAC-PL, or nonimmune mouse serum at 1:100 dilution ratio and rabbit monoclonal anti-alpha tubulin (acetyl K40) antibody (ab179484, Abcam, Cambridge, UK) at 1:500 dilution ratio in the blocking buffer for 1 h. After washing with PBS, the slides were incubated with goat anti-mouse IgG (Alexa 555; Molecular Probes, Eugene, OR, USA) and goat anti-rabbit IgG (Alexa 488; Molecular Probes) at 1:1000 dilution ratio in the blocking buffer for 1 h. After washing with PBS, samples were mounted in Slow Fade^TM^ Gold antifade reagent (Invitrogen, Waltham, MA, USA). The slides were observed under a fluorescence microscope (Elyra7; Zeiss, Oberkochen, Germany) with 40× objective and processed with apotome.

### 2.5. Analysis of Flagellar Waveforms in Sperm Chemotaxis

Sperm chemotaxis was examined as described previously [13,17]. Briefly, semen were suspended in 2000 volumes of ASW with 0.5% DMSO or 10 μM KH-7. After incubation for 3 min, sperm were activated using 1 mM theophylline and then placed in the observation chamber. Sperm movement around the micropipette tip containing 1 μM SAAF was observed under a phase contrast microscope (BX51, Olympus, Tokyo, Japan) with a high-speed CCD camera (HAS-D3, DITECT, Tokyo, Japan) with 20× objective (UPlanApo, Olympus). Flagellar waveforms were captured using a white-LED (W42182-U, Seoul Semiconductor, Gyeonggi-do, Korea) and a laboratory-made LED stroboscopic illumination system synchronized with the exposure signals from the high-speed camera. Images were taken at a frame rate of 200 fps with 0.05 msec pulse from the LED. The flagellar waveforms and the flagellar curvature were analyzed using Bohboh software (Bohboh 4, Rel.4.88, Bohbohsoft, Tokyo, Japan). Individual images of sperm flagella were tracked automatically, and their curvatures were calculated based on the method of Baba and Mogami [21]. The asymmetric index was obtained from the ratio of maximal curvature of the P-bend and that of the R-bend as described by Mizuno et al. [22].

### 2.6. Imaging Analysis of [Ca^2+^]_i_

For Ca^2+^ imaging, Fluo-8H AM (AAT Bioquest, Sunnyvale, CA, USA) was used as a fluorescent probe. The dye-loaded sperm were prepared as described previously [23]. Fluorescent images of the sperm were observed with a microscope (IX71, Olympus), and captured on a PC connected to a digital CCD camera (ImagEM, C9100-13; Hamamatsu Photonics, Hamamatsu, Japan) at 50 frames/s using Aquacosmos (Hamamatsu Photonics). For fluorescence illumination, a stroboscopic lighting system with a power LED was used as described by Mizuno et al. [22]. Fluorescent signal intensity and sperm swimming trajectories were analyzed using the Bohboh software.

### 2.7. Statistical Analysis

All experiments were repeated at least three times with three different animals. Statistical significance was calculated using Student’s *t*-test; *p*  <  0.05 was considered significant. The data were analyzed using the R script.

## 3. Results

### 3.1. Expression of Full-Length and Truncated sAC in Ciona Sperm

The amino acid sequence of *Ciona* sAC (Ci-sAC) has two catalytic domains, a P-loop motif that is characteristic of the AAA superfamily, and a tetratricopeptide repeat (TPR) domain, as similarly seen in the sAC of sea urchins and mammals [7] (Figure 1A). Expression of both full-length and truncated sAC proteins have been reported in mammals, sea urchins and corals [3,5,24]. To confirm the presence of sAC protein isoforms in *Ciona* sperm, we prepared polyclonal antibodies against bacteria-expressed polypeptides of two regions in Ci-sAC (sAC-C1 and sAC-PL) as shown in Figure 1A. Expression of sAC protein isoforms in *Ciona* sperm was confirmed using Western blotting. The antibody against the region sAC-C1 containing catalytic domain C1 specifically recognized three protein bands in the whole *Ciona* sperm proteins. On the other hand, another antibody against the region sAC-PL recognized a single protein band (Figure 1B). The largest molecular mass of the band recognized by both antibodies was estimated as 200 kDa, which agreed with the mass predicted for Ci-sAC full-length protein (206.9 kDa). The size of truncated isoforms only recognized by the antibody sAC-C1 was estimated as 53 kDa and 45 kDa, respectively. These suggest that both full-length and truncated sAC proteins are expressed in *Ciona* sperm. Immunofluorescent microscopy clearly showed that Ci-sAC was localized along the sperm flagellum (Figure 1C). Interestingly, the sAC-C1 antibody recognized all isoforms stained along the entire length of the sperm flagellum and the sAC-PL antibody recognized only full-length isoform stained the proximal most region of the flagellum. Specific localization of the sAC isoform suggests that the sAC proteins might change their localization after truncation.

### 3.2. sAC Is Involved in Turn Movement during Sperm Chemotaxis 

We previously reported that *Ciona* sperm chemotaxis is significantly suppressed by the sAC inhibitor [13]. To elucidate the role of sAC in sperm motility regulation, we analyzed the changes in sperm flagellar waveforms during chemotaxis. Around the tip of glass capillary containing the SAAF with 1% agar, *Ciona* sperm showed the characteristic turn movement consisting of a weak asymmetric wave to a strong asymmetric wave, followed by a symmetric wave (Figure 2A; also see [11,12,13,17]). These sequential changes in flagellar bending induced the efficient turn toward the attractant source in sperm chemotaxis. The asymmetry index, which is a parameter of flagellar asymmetry calculated from the flagellar curvature, increased slowly and then decreased (Figure 2B,C). In sperm treated with a sAC inhibitor KH-7, the gradient of the attractant concentration induced a conversion from a weak asymmetric wave to a strong asymmetric and then symmetric wave instantaneously. However, both waveforms were not stable and quickly returned to a weak asymmetric state (Figure 2D). The asymmetry index fluctuated compared with the control, resulting in zigzag swimming trajectories and insufficient chemotaxis turns (Figure 2E,F). The maximum asymmetric index during turn movement was significantly increased in KH-7-treated sperm (Figure 3). Asymmetry duration was significantly shorter, and the number of asymmetries per turn was significantly increased in KH-7-treated sperm. These results indicate that sAC inhibition destabilizes the asymmetric waveform. 

### 3.3. sAC Is Involved in the Dynamics of [Ca^2+^]_i_ during Sperm Chemotaxis 

Next, we focused on the dynamics of intraflagellar calcium, which plays an important role in the asymmetrization of flagellar waves. As we previously reported [13], the sAC inhibitor induced a decrease in swimming velocity and circular path. We could not detect significant differences in the basal [Ca^2+^] levels in flagella between the 0.5% DMSO-treated control and the sAC inhibitor-treated sperm. The average of F/F0- in KH-7-treated sperm before the insertion of the glass needle containing the SAAF was 0.91 ± 0.18. The effect of the sAC inhibitor on the [Ca^2+^] dynamics was much more pronounced in chemotaxis than in activation with the SAAF. In the 0.5% DMSO-treated control sperm, the intraflagellar calcium concentration transiently increased, inducing a strong asymmetric wave and sequential turn movement (Figure 4A,B). On the other hand, KH-7-treated sperm showed a strong and prolonged increase in the intraflagellar calcium concentration in the gradient of the sperm attractant (Figure 4A,B). Both the mean and maximum intraflagellar calcium concentrations in KH-7-treated sperm were significantly increased compared to control sperm (Figure 4C). The time to peak and the rate of increase in the calcium concentration also showed higher values in KH-7-treated sperm; however, there was no significant difference in the rate of decrease in the calcium concentration (Figure 4C). Nonetheless, intraflagellar [Ca^2+^] remained high due to the excessive entry of Ca^2+^ (Figure 4B). These results suggest that sAC affects the suppression of the excess calcium influx during chemotaxis.

## 4. Discussion

In this study, we have shown that the sAC protein is present in *Ciona* sperm flagella as multiple isoforms. The structure of *Ciona* sAC (Ci-sAC) is similar to that reported for mammals, sea urchins and corals [3,7]. The presence of full-length and truncated forms is also a feature shared with other organisms. In *Ciona* sperm, the full-length sAC (sAC_fl_) is localized only in the proximal region of the flagella, whereas the truncated sAC (sAC_tr_) is present throughout the entire flagellum. It is possible that the full-length form is anchored to other structures in the proximal region through the TPR domain. In contrast, the truncated form is free, unbound to other structures and is distributed evenly along the flagellum. The full-length form is known to show autoinhibitory activity [25], suggesting a localized cAMP production along a flagellum. In sea urchins, sAC_fl_ and sAC_tr_ are localized at the flagella and the head, respectively, suggesting that sAC has multiple roles in sperm function, such as flagellar motility and acrosomal reaction [24]. On the other hand, sAC is localized toward the midpiece, and sAC_tr_ is highly expressed in mature sperm in mice [5,9]. sAC truncation has been previously reported to play a role in the regulation of the enzymatic activity of sAC. It is known that in mammalian sAC, the truncated form is more active than the full-length form and that the full-length form suppresses the activity of the truncated form [4,25,26]. The expression pattern and the localization of sAC isoforms in *Ciona* is more similar to mice sperm than the sperm of sea urchins. It is interesting for a clarification to be published in the future explaining how the truncated form of sAC is generated and how it regulates the motility activation and sperm chemotaxis.

Asymmetric waves contribute to form swimming trajectories with a small circle, while symmetric waves contribute to straight swimming. During chemotaxis, sperm change their swimming direction by exhibiting these two waveforms in sequence, resulting in a chemotactic turn, as they swim away from the attractant source (Figure 5). We have previously reported that sAC inhibition blocks this turn movement [13]. In this study, we analyzed the flagellar waveforms during chemotaxis in detail to clarify the role of sAC in the regulation of flagellar motility. We found that sAC inhibition does not directly suppress the waveform conversion but significantly increases the maximum asymmetric index and shortens the duration of asymmetric waves. As a consequence, the waveform becomes too unstable to resume a normal symmetric waveform. This causes a zig-zag swimming trajectory and thus leads to the failure of the chemotactic turn. Visualization of intraflagellar Ca^2+^ revealed that the inhibition of sAC resulted in an excessive calcium influx during chemotaxis; therefore, a strong and persistent asymmetry wave was formed, and the completion of the calcium efflux was also prolonged. These results suggest that sAC is not directly involved in triggering the waveform conversion but is instead involved in maintaining the attractant-induced waveform change by regulating the calcium efflux. To form an efficient turn to approach the chemoattractant source, the [Ca^2+^]_i_ increase along the asymmetric wave should be transient and end with a subsequent Ca^2+^ efflux. The regulation of the sAC and Ca^2+^ efflux in response to the SAAF-induced [Ca^2+^]_i_ increase would be responsible for the intracellular signaling of chemotaxis (Figure 5). Sperm chemotaxis in *Ciona* is caused by a transient calcium influx into flagella [23]. The high efflux of calcium through the plasma membrane Ca^2+^-ATPase PMCA, the receptor of the SAAF [27], keeps the flagellar waveform weakly symmetric (Figure 5, ‘Steady’). When sperm swims away from the source of attraction, the binding of the SAAF to the PMCA decreases, stopping the calcium efflux. This makes the calcium influx through calcium channels dominant and increases [Ca^2+^]_i_ (Figure 5, ‘Turn’). The change in the swimming direction toward the attractant makes the SAAF bind to the PMCA again, resuming the calcium efflux and resulting in the formation of symmetric waveforms that instruct sperm to swim straight (Figure 5, ‘Straight’). 

This study suggests that sAC controls flagellar waveforms during sperm chemotaxis by regulating the calcium efflux. Sperm sAC is activated with calcium [4], which is increased by the suppression of the PMCA via release of the bound SAAF in the ‘Turning’ phase. It is possible that the cAMP produced by sAC activates cAMP-dependent protein kinase (PKA) to phosphorylate 14-3-3ε, which is one of the major phosphorylated proteins in *Ciona* sperm [28]. In fact, a 14-3-3ε binding site is present at the N-terminal region of the PMCA [22]. cAMP is likely to regulate the membrane potential by acting on tetraKCNG and HCN channels [13], which causes membrane hyperpolarization and could activate NCX [29] to promote the calcium efflux (Figure 5). Calcium ions interact with calaxin, which is the calcium sensor of outer arm dynein and controls the formation and propagation of asymmetric flagellar waveform formation and propagation [22,30]. cAMP may also be involved in flagellar waveform regulation via the phosphorylation of axonemal proteins. To elucidate the mechanism of sAC-mediated regulation of sperm chemotaxis, it is essential to characterize proteins that are phosphorylated in a cAMP-dependent manner.

## 5. Conclusions

We found that both full-length and truncated isoforms of the sAC protein are expressed and localized in *Ciona* sperm flagella. Analysis of sperm flagellar motility and the dynamics of [Ca^2+^]_i_ revealed that sAC plays a key role in sperm chemotaxis, possibly by regulating the clearance of [Ca^2+^]_i_ and by modulating Ca^2+^-dependent flagellar waveform conversion. 

## Figures and Tables

**Figure 1 biomolecules-13-01594-f001:**
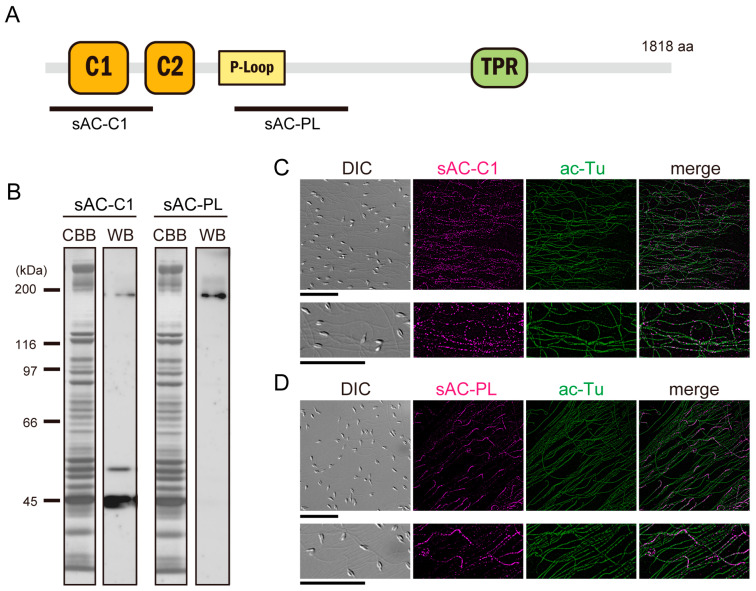
(**A**) Schematic representation of the Ci-sAC. The bold line shows the region used for the antigen to raise a polyclonal antibody. (**B**) Western blots of whole sperm protein with the antibodies against the two regions of Ci-sAC. CBB-stained pattern of whole sperm proteins (**left**) and corresponding immunoblots (**right**) are shown. Original images of (**B**) can be found in Appendix A. (**C**,**D**) Immunolocalization of Ci-sAC in sperm with the antibodies against the two protein regions. The differential interference contrast (DIC) image, Ci-sAC (magenta), alpha-tubulin, acetylated (green) and the merged image are shown. Scale bar: 20 µm.

**Figure 2 biomolecules-13-01594-f002:**
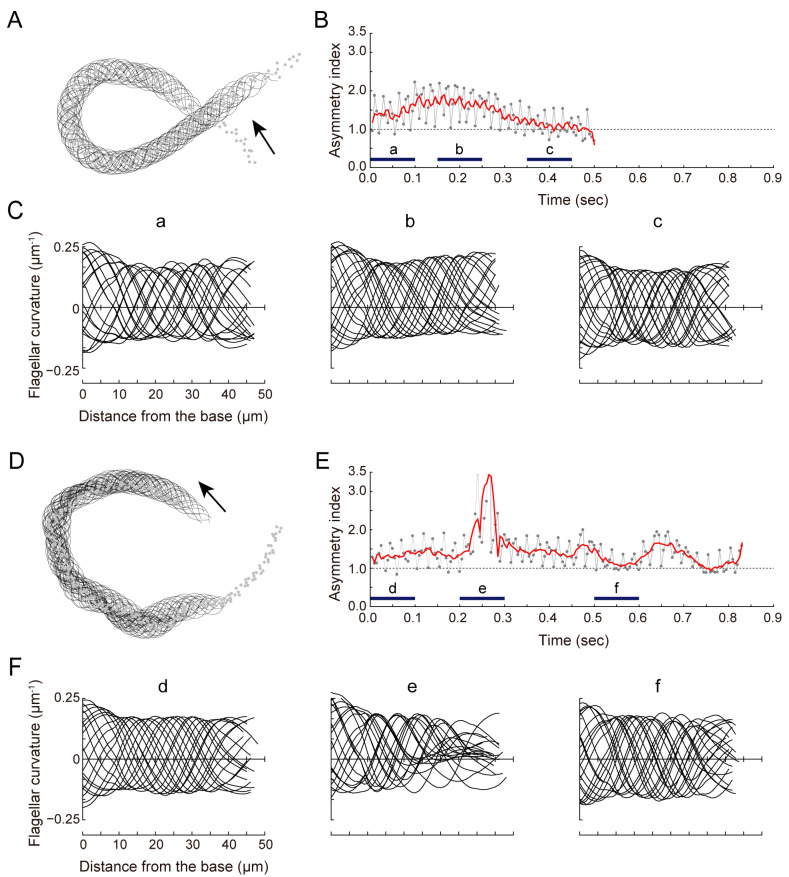
Effects of a sAC inhibitor KH-7 on the flagellar motility in *Ciona* sperm during chemotactic turn. Data that are shown comprise flagellar bending patterns (**A**,**D**), changes in asymmetric index (**B**,**E**) and flagellar curvature (**C**,**F**) during the chemotactic turn of sperm treated with 0.5% DMSO (control, **A**–**C**) and sperm treated with 10 μM KH-7 (**D**–**F**). (**A**,**D**) Flagellar bending patterns. Data from 100 (**A**) and 162 (**D**) waveforms are overwritten at 5 msec intervals. Dots and arrows indicate head position and direction of movement, respectively. (**B**,**E**) Changes in asymmetric index. Asymmetric index was calculated as the ratio of maximal curvatures of both bends (P-bend_Max_/R-bend_Max_). A raw value from each waveform (gray dot) and a smoothened value (red line) obtained using the Savitzky–Golay method (9 points) are shown above. (**C**,**F**) Flagellar curvature is plotted against the distance from the base of flagellum. Data from 20 waveforms from a–c (**B**) or d–f (**E**) are separately overwritten in (**C**) or (**F**), respectively.

**Figure 3 biomolecules-13-01594-f003:**
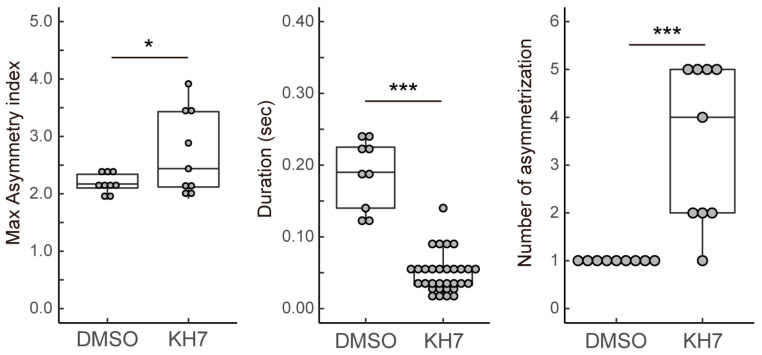
Comparison of the maximal asymmetric index, duration of asymmetric wave formation and number of asymmetries during one turn between *Ciona* sperm treated with 0.5% DMSO (control) and 10 μM KH-7 (KH7). These parameters for wave asymmetry were estimated from the comparison between the P- and R-bend. Asymmetric index > 1.5 is defined as an asymmetric wave. Distribution of values is plotted in a box plot. *** Significant at *p* < 0.001, * *p* < 0.05 (Student’s *t*-test) in comparison to the control.

**Figure 4 biomolecules-13-01594-f004:**
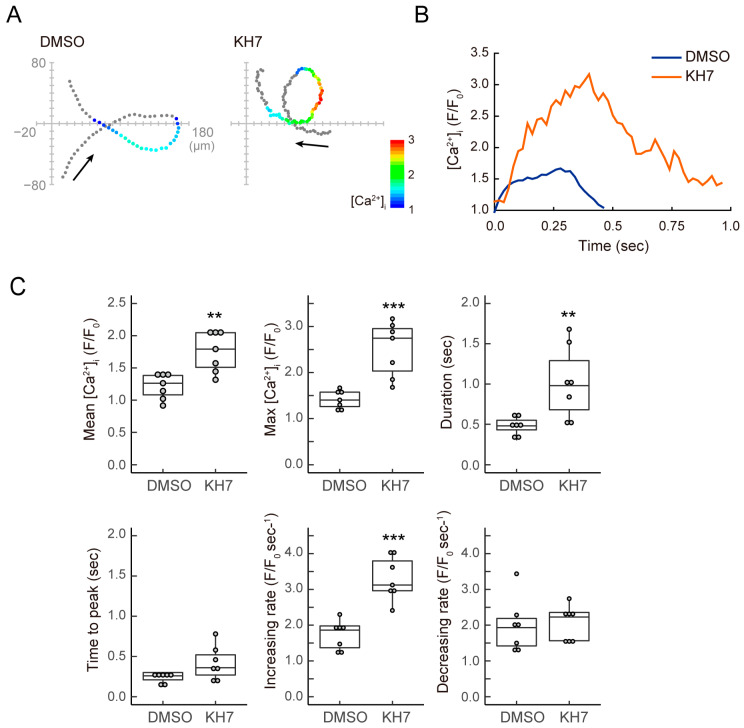
Effects of a sAC inhibitor KH-7 on [Ca^2+^]_i_ dynamics during sperm chemotaxis. (**A**,**B**) Trajectories of the sperm head (**A**) and changes in [Ca^2+^]_i_ signals yielded by the sperm tail (**B**) in *Ciona* sperm treated with 0.5% DMSO (control; DMSO) and 10 μM KH-7 (KH7) around the tip of a capillary containing 1 μM SAAF. In image (**A**), the origin of the coordinates indicates the capillary tip. The arrows indicate the swimming direction of the sperm. The dots show the head position with the color representing the average intensity of [Ca^2+^]_i_ signals in pseudo colors. The color scale represents the lookup table (LUT) for fluorescence signals. (**C**) Comparisons of [Ca^2+^]_i_ dynamics between *Ciona* sperm treated with 0.5% DMSO (control; DMSO) and 10 μM KH-7 (KH7). Data represent the mean and maximum of [Ca^2+^]_i_, duration of [Ca^2+^]_i_ increase, time to the peak of [Ca^2+^]_i_ and the increasing and decreasing rate of [Ca^2+^]_I_ in the sperm around the tip of a capillary containing 1 μM SAAF. [Ca^2+^]_i_ is expressed as F/F0, which is the value of the fluorescent intensity from the whole flagella (F) divided by the average intensity of basal [Ca^2+^]_i_ before SAAF stimulation of the whole flagella (F0). Distribution of values is plotted in a box plot. *** Significant at *p* < 0.001, ** *p* < 0.01 (Student’s *t*-test) in comparison to the control.

**Figure 5 biomolecules-13-01594-f005:**
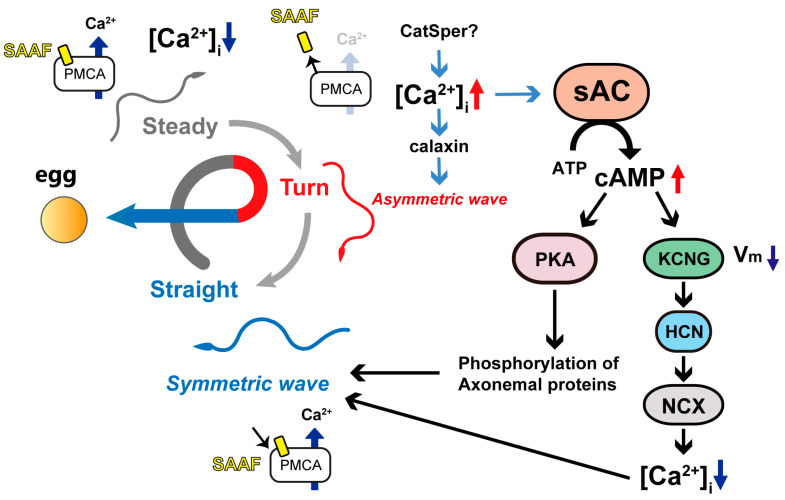
Roles of sAC in the turn movement of *Ciona* sperm. The turn movement toward the egg consists of a weakly asymmetric wave (steady phase) to a highly asymmetric wave (turn phase), followed by a symmetric wave (straight swimming phase). 1. *Steady state*. While the SAAFs are bound to the plasma membrane Ca^2+^-ATPase (PMCA), which pump Ca^2+^ out from the cytoplasm. [Ca^2+^]_i_ is kept at a constant balance with the calcium efflux and a calcium channel, such as CatSper. 2. *Turn state*. When the SAAF dissociates from the PMCA, the Ca^2+^ efflux caused by the PMCA ceases and the calcium influx becomes dominant, inducing the increase in [Ca^2+^]_i_. The increase in [Ca^2+^]_i_ induces an asymmetric wave through calaxin. Ca^2+^ simultaneously activates sAC and promotes cAMP synthesis. cAMP induces hyperpolarization by activating the tetrameric, cyclic nucleotide-gated, K+-selective channel (tetraKCNG). cAMP and hyperpolarization activate the hyperpolarization-activated and cyclic nucleotide-gated channel (HCN). Hyperpolarization also promotes the calcium efflux from the Na^+^/Ca^2+^ exchanger (NCX). *3. Straight swimming phase.* In addition to the sAC-mediated Ca^2+^, the SAAF recruits PMCA again, resulting in a decrease in [Ca^2+^]_i_ and the formation of symmetric waves. cAMP also induces phosphorylation and dephosphorylation of axonemal proteins via the cAMP-dependent protein kinase (PKA), which causes the change in sperm’s flagellar waveform.

## Data Availability

All data presented in this study are available in the article.

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
