# Peer review of "The Role of Soluble Adenylyl Cyclase in the Regulation of Flagellar Motility in Ascidian Sperm"

_biomolecules, 2023, doi:10.3390/biom13111594_

Round 1

Reviewer 1 Report

Soluble adenylyl cyclase is a crucial enzyme for sperm motility regulation in the vertebrates and the invertebrates although various species such as zebrafish and fruit fly do not possess (lost) the gene encoding this protein. The authors previously reported that ascidian (C. intestinalis) sperm possess both transmembrane adenylyl cyclase  (mAC) and soluble adenylyl cyclase (sAC) and sAC is crucial for the sperm chemotaxis toward SAAF rather than the motility initiation. In this study, the authors reported further characterization of the sAC of C. intestinalis spermatozoa. Firstly, they demonstrated the presence of two forms of sAC, full-length of sAC (sACfl) and truncated form of (sACt), by Western Blotting using two antibodies generated with two antigens, anti-P-loop (sAC-PL) found only in sACfl and anti-catalytic domain (sAC-C1) found in both sACfl and sACt. Furthermore, they demonstrated by immunofluorescence that both forms of sAC are localized in sperm tail, but the distributions of two forms are not completely the same. Subsequently, the authors showed results of extensive analyses of effects of specific inhibitor of sAC, KH7, on sperm flagellar bend and [Ca2+]i during sperm chemotaxis. Although SAAF signaling cascade to explain all sperm behavior (motility initiation and chemotaxis) is complicated and an issue to debate, the authors proposed a model to explain the role of sAC in sperm chemotaxis towards SAAF.

Major comments:

In mouse, a major catalytic activity of sAC is likely to be depending on full length (sACfl)  (Wang et al., 2007) although the Western blotting indicates that sACfl is a minor form of sAC in mouse sperm. On the other hand, in sea urchin, most of sAC was found as sACfl in Western blotting using anti-catalytic domain antibody (Nomura et al., 2006). In this study, Western blotting with sAC-C1 antibody indicates that truncated form of sAC (sACt) is dominant in C. intestinalis spermatozoa. In this sense, sAC of C. intestinalis sperm is similar to sAC of mammals (mouse). The authors should discuss these points. Image of immunofluorescence of sAC-C1 and sAC-PL shows a clear difference of localization of two forms and supports that sACt is dominant because sAC-C1 detects both forms of sAC and should not exhibit a different pattern with sAC-PL unless sAC-C1 is dominant. However, it is difficult to recognize the detailed individual sperm images because image of each cell is too small. Therefore, I recommend the authors to add enlarged representative sperm images together with the current images.

Figure 2E clearly demonstrated zig-zag trajectories of C. intestinalis spermatozoa during chemotaxis in the presence of sAC-specific inhibitors as one of prominent features of this inhibitor. One of the important questions is that if this fluctuation of flagellar form is related to [Ca2+]i levels (rates of [Ca2+]i changes rather than absolute [Ca2+]i level) or not. Changes in [Ca2+]i shown in Figure 4B in the presence of KH7 show some fluctuations after a main increase. Are they significant? related to asymmetry index of flagellar forms? It would be quite interesting if the authors find any correlation between [Ca2+]i changes and asymmetry index.

According to the title and Abstract of the paper, the main theme of this work is the role of sAC in C. intestinal sperm chemotaxis. However, the authors proposed SAAF signaling in Figure 5 basically based on the role of Ca2+ rather than sAC (or cAMP). Therefore, principal message of this paper is not so clear. Actually, molecular mechanisms how cAMP affects the [Ca2+]i are all speculative without experimental evidence. Therefore, I suggest modifying the figure, including omission of this model.  

The [Ca2+]i in this work is expressed by F/F0. According to the cited reference (Shiba et al., 2008), “[Ca2+]i was expressed as F/F0, which is the value of the fluorescent signals from heads or flagella (F) divided by the minimum value of signals emitted by the heads (F0)”. It is possible to quantitatively compare F/F0 values if F0 value is constant. However, if KH7 alters F0 value, F/F0 is not appropriate to use. Actually, the authors reported previously (Fig. 7 of Shiba and Inaba 2014) that KH7 alters the curvature of sperm swimming trajectories, which suggests an increase in the resting [Ca2+]i. Actually, the model of this work is that cAMP produced by sAC positively regulates the cytosolic Ca2+ clearance. Therefore, in order to compare the [Ca2+]i in the presence or absence of KH7, simple fluorescence intensities rather than normalized values with F0 might be better. The use of F/F0 could sub-estimate the [Ca2+]i in the presence of KH7 although the principal conclusion should not be changed.

Minor points:

In this work, the term “CNG channel” is used as a family of channels regulated by cyclic nucleotides as the previous paper (Shiba and Inaba, 2022). However, CNG channels represent a subfamily of channels regulated by cyclic nucleotides. Therefore, I recommend the authors to replace CNG by another term to avoid any confusion. Cyclic nucleotide-regulated channels (CNR channels) or cyclic nucleotide-modulated channels (CNM channels) are a few of options.

In the text, the authors use LTU without definition. Please define it.

Author Response

We thank the reviewer for these helpful comments. We have responded to the reviewers' comments as the attached file.

Reviewer 2 Report

In the manuscript “The role of soluble adenylyl cyclase in the regulation of flagellar motility in ascidian sperm” the authors propose to examine and discuss the role of sAC in flagellar motility of ascidian C. intestinalis under the effect of Ca+.  The aim is interesting, the manuscript is easy to read, the methods are clear, the results well-presented and properly discussed in this section.  I do not have any special concerns regarding the material presented in this manuscript.

Author Response

We thank the reviewer for reading our manuscript.

Reviewer 3 Report

The manuscript entitled “The role of soluble adenylyl cyclase in the regulation of flagellar motility in ascidian sperm” by Shiba and Inaba is intended to show the role of sAC the dynamics of [Ca2+]i of ascidian sperm motility. The manuscript is interesting; however, some issues should be pointed out.

1.- Please place every figure in the manuscript when first mentioned.

2.- Figure 1. Please, capture more pictures with a different microscope objective (100x) to have a precise view of the sperm localization of the sAC, it is not clear with that zoom.

3.- Please, change from intracellular Ca2+ to [Ca2+]i in all text.

4.- There is a lack of a negative control in the intracellular Ca2+ dynamics during sperm chemotaxis experiment, maybe you can use a selective CatSper inhibitor, since is mentioned on Figure 5. 

5.- I feel the work is a little short, and it will be helpful to the final conclusion to evaluate the [cAMP]i in presence of SAAF and KH7 as a negative control.

Author Response

(The authors gave the same response as above.)

Round 2

Reviewer 3 Report

I have no further comments; however, just note that figure 1 is repeated on page 8.

Author Response

Thank you for reviewing  our manuscript and helpful comments. We will delete the overlapped figure in next revision.